# Virulence-Inhibiting Herbal Compound Falcarindiol Significantly Reduced Mortality in Mice Infected with *Pseudomonas aeruginosa*

**DOI:** 10.3390/antibiotics9030136

**Published:** 2020-03-24

**Authors:** Pansong Zhang, Qiaolian Wu, Lin Chen, Kangmin Duan

**Affiliations:** 1Key Laboratory of Resource Biology and Biotechnology in Western China, Ministry of Education, Northwest University, Xi’an 710069, China; zps102065106@stumail.nwu.edu.cn (P.Z.); wqlian0805@gmail.com (Q.W.); 2Department of Oral Biology & Medical Microbiology & Infectious Diseases, Rady Faculty of Health Sciences, University of Manitoba, 780 Bannatyne Ave., Winnipeg, MB R3E 0W2, Canada

**Keywords:** *Pseudomonas aeruginosa*, falcarindiol, anti-virulence therapeutic, type III secretion system, antibiotics, quorum sensing

## Abstract

Antipathogenic compounds that target the virulence of pathogenic bacteria rather than their viability offer a promising alternative approach to treat infectious diseases. Using extracts from 30 Chinese herbs that are known for treating symptoms resembling infections, we identified an active compound falcarindiol from *Notopterygium incisum* Ting ex H. T. Chang that showed potent inhibitory activities against *Pseudomonas aeruginosa* multiple virulence factors. Falcarindiol significantly repressed virulence-related genes, including the type III secretion system (T3SS); quorum sensing synthase genes *lasIR* and *rhlIR*; *lasB*; motility-related genes *fliC* and *fliG*; and phenazine synthesis genes *phzA1* and *phzA2. P. aeruginosa* swarming motility and pyocyanin production were reduced significantly. In a burned mouse model, falcarindiol treatment significantly reduced the mortality in mice infected with *P. aeruginosa,* indicating that falcarindiol is a promising antipathogenic drug candidate for treating *P. aeruginosa* infections.

## 1. Introduction

Antibiotic resistance in pathogenic bacteria has become one of the biggest threats to human health that we face today. Antibiotic resistance not only threatens to severely limit our ability to treat infectious diseases but also our ability to perform various medical procedures, such as surgeries, tissue transplants, and cancer treatments [1]. All these procedures require preventing bacterial infections and the use of antibiotics are the only option. There is an urgent need to devise new therapeutic options for the treatment of infections caused by drug-resistant pathogens. Finding new antibiotics is one of the top health research priorities identified by the World Health Organization. Unfortunately, even though we are rapidly running out of effective antibiotics to treat infections, particularly for Gram-negative bacteria, the search for new and potent antibiotics has been unsuccessful to date. Therefore, innovative approaches are needed to find new and effective antimicrobial therapeutics.

Traditional antibiotic gents that target bacterial viability have been investigated for many years. However, resistance to such antibiotics that target pathogens’ viability emerges quickly due to the rise of mutations or transfer of resistance determinants, making the newly developed drugs rapidly lose their efficacy and usefulness. One of the promising new approaches for anti-infective therapeutics is to develop antibacterial agents that target the virulence—so-called antipathogenics [2]. Virulence factors such as toxins, cytolysins, or proteases enable bacterial pathogens to cause damage to the host or evade the host immune system to result in disease. In many Gram-negative bacteria, the importance of virulence factors such as the type III secretion system (T3SS) and quorum sensing (QS) systems in bacterial pathogenicity and disease progression has been documented in a great deal of studies [3,4,5,6,7,8]. Antivirulence therapeutics might not only prevent the damage to the host but also preserve the host endogenous microbiome. Theoretically, such therapeutics would exert less selective pressure, and hence would be less likely to cause drug resistance, because they do not directly target the viability of the pathogen. Therefore, antipathogenics represent one of most promising classes of novel antibiotics.

*Pseudomonas aeruginosa*, one of the predominant nosocomial pathogens, can cause a range of infections in humans, including pneumonia and urinary tract infection, bloodstream infection, infections in burn patients, and pulmonary infections in patients with cystic fibrosis. Drug resistant in *P. aeruginosa*, including carpenapenin-resistant strains, has caused enormous difficulties in clinical treatment of infections. The successful infection of *P. aeruginosa* in diverse hosts is due to the profusion and diversity of virulence factors such as Type III secretion systems (T3SS), phenazine compound production, proteases secretion, and biofilm formation. T3SS plays an important role in *P. aeruginosa* acute infection and functions by injecting effector proteins into the cytoplasm of eukaryotic host cells [7,8,9,10]. The individual virulence factors, such as T3SS, serve as promising targets for antivirulence therapy. 

Global regulatory systems, such as QS systems and GacS/GacA (two-component regulatory system), have been documented to play significant roles in modulating the expression of diverse virulence factors [11,12]. Numerous cellular and secreted virulence factors are coordinately regulated by the quorum sensing global regulatory system [13]. There are three intertwined QS systems in *P. aeruginosa*: the two acyl homoserine lactone (AHL)-mediated *las* and *rhl* systems, and the 2-alkyl-4 (1H)-quinolone (AHQ) signal-based system [14,15]. Global regulatory systems that control virulence factors or the pathogenicity of the pathogens constitute another promising target for developing antipathogenics. An example is a QS inhibitor from a halogenated furanone compound isolated from a marine alga *Delisea pulchra* [16,17]

Chinese herbs have a long history of use in treating infectious diseases. Many components from herbs have been identified as effective in the treatment of human disease [18,19,20]. However, the exact mechanism is often not understood. Based on the relevance of these medicinal materials in treatment of infection, they may represent a potentially rich resource of chemical diversity for identification and exploration of antivirulence compounds. 

Here, we report the identification and characterization of the natural compound falcarindiol from the chinese medicinal plant *Notopterygium incisum* as a potential antipathogenic agent. We present data showing that falcarindiol significantly inhibited a broad range of important virulence factors in *P. aeruginosa*, including the T3SS, swarming motility, and the production of pyocyanine. Treatment with falcarindiol significantly reduced the mortality in mice infected with *P. aeruginosa,* supporting falcarindiol as a promising antipathogenic drug candidate. 

## 2. Results and Discussion

### 2.1. The Effect of Crude Extract of Notopterygium Incisum Ting ex H. T. Chang on Virulence Factors in P. aeruginosa

In the initial screening for virulence inhibitors, we tested the effect of 30 herbal medicines known for the functions of “Qing Re Jie Du” [21], a term equivalent to “lowering fever and alleviating toxicity” (i.e., treating symptoms resembling infections), on virulence-associated gene expression in *P. aeruginosa*. The gene expression was monitored as light production from the promoterless *luxCDABE* operon downstream of 14 promoters of the virulence factor genes or operons (Table 1) [22,23,24,25]. Among these samples, the crude extract from *Notopterygium incisum* showed the most potent inhibition of multiple virulence factors and quorum sensing systems. The ethanolic extract of *Notopterygium incisum* inhibited the expression of *lasI, rhlI, lasB, fliC, phA1, phzA2, exoS, rhlR,* and *lasR*. The inhibition of *rhlI* and *phzA1* by the extract of *Notopterygium incisum* is shown in Figure 1. 

The results indicate that there are bioactive compounds in the *Notopterygium incisum* extract that could inhibit the transcription of genes involved in the virulence of *P. aeruginosa*. In order to exclude the possibility that the inhibition of the virulence factors was due to the growth inhibition, the colony formation in the plates with or without different concentrations of the crude extract of *Notopterygium incisum* was compared. As seen in Figure 1C, there were no significant changes in CFU counts between the control and the different treatments, indicating no inhibition of *P. aeruginosa* growth by the crude extract at 2 mg/mL.

Encouraged by the results, we initially used the *Drosophila melanogaster* infection model to test the effect of the *Notopterygium incisum* extract on the mortality of the infected fruit flies. Such an infection model is economical and easy to perform, and has been widely used to assess *P. aeruginosa* pathogenicity [28]. As shown in Figure 2, the presence of *Notopterygium incisum* extract significantly decreased the fly mortality of *P. aeruginosa* infection. The result suggests that the presence of *Notopterygium incisum* extract attenuated the pathogenicity of *P. aeruginosa*.

### 2.2. Inhibition of Virulence-Associated Genes in P. aeruginosa and Phenotypes by Falcarindiol

The herbs used in our study have been investigated intensively in recent years and many bioactive compounds have already been identified [29,30,31]; therefore, as the first step, we took the strategy of testing the known compounds available to identify the bioactive compounds in *Notopterygium incisum*. The next approach was to isolate individual compounds from the extract if needed. One of the several major compounds from the plant *Notopterygium incisum* K. C. Ting ex H. T. Chang is falcarindiol a C(17)-polyacetylene [32], which, providentially, was found to be very active against the expression of the virulence factors tested, except *aprA.* It had no influence on the growth of *P. aeruginosa* PAO1 at a concentration up to 64 µg/mL.

As shown in Figure 3, falcarindiol significantly inhibited the expression of a range of important virulence factors in *P. aeruginosa*, such as T3SS genes *exoS*, *exoT*, and *exsC*; quorum sensing regulator genes *lasI*, *rhlI*, and *lasB*; motility-related gene *fliC*; and phenazine synthesis gene operons *phzA1* and *phzA2*; whereas the expression of *aprA*-encoding alkaline metalloproteinase precursor was not changed. Other genes affected by falcarindiol include *fliG*, *exsD*, *rhlR*, and *lasR*.

It is interesting to note that falcarindiol is a typical constituent of roots and rhizomes of Apiaceae plants used worldwide, including in Europe, such as herbs and the commonly used vegetables carrot (*Daucus carota L*.) [33] and celery (*Apium graveolens L*.) [34], among others. This may suggest that some traditional food ingredients may play a role in preventing diseases in humans.

In *P. aeruginosa*, the biosynthesis of phenazine compounds is carried out by enzymes encoded by two seven-gene operons, *phz1 (phzA1B1C1D1E1F1G1)* and *phz2 (phzA2B2C2D2E2F2G2)* [35]. The results above showed that falcarindiol inhibited the expression of *phzA1* and *phzA2*. Thus, we examined the effect of falcarindiol on the production of pyocyanin, the major phenazine compound in *P. aeruginosa*. In accordance with the decreased gene expression, falcarindiol significantly reduced the production of pyocyanine at the concentration of 32 μg/mL (Figure 4A). To further verify the effect of falcarindiol, we also tested the swarming motility of *P. aeruginosa* PAO1 in the presence of falcarindiol. As shown in Figure 4B, the presence of falcarindiol (32 μg/mL) impaired swarming motility, consistent with the decreased expression of flagella-related genes caused by falcarindiol.

### 2.3. Falcarindiol Attenuates the Pathogenicity of P. aeruginosa PAO1 in a Burned Mouse Model

The experimental animals consisted of female C57BL/6J mice (weight, 20 g ± 2 g). The burned mouse model was adopted to analyze *P. aeruginosa* pathogenicity in vivo and test falcarindiol’s effect on *P. aeruginosa* pathogenicity. In the initial experiment, two groups of mice (*n* = 10 per group) were burned and inoculated with 5 × 10^7^ CFU of *P. aeruginosa*. The treatments with falcarindiol at 10 mg per kg body weight injected intraperitoneally started at 12 h postinfection, and the survival of the mice was recorded every day for a total of 6 days. As shown in Figure 5A, the mortality of the mice without falcarindiol treatment reached 100% (0% survival rate) after 4 days. In contrast, the survival rate of the mice with the falcarindiol treatment was 90% after 4 days.

Based on the preliminary data, the second animal study was carried out with a lower infection dose, two different dosages for drug treatment, and a slightly large sample size (*n* = 12 per group). The adjusted infection dose was 5 × 10^5^ CFU of *P. aeruginosa* and the survival of the mice was recorded every day for a total of 8 days. As shown in Figure 5B, the final survival rate of the mice without falcarindiol treatment (saline-treated) was 33.3%. The group treated with a low dose of falcarindiol had a significantly higher final survival rate of 66.7%, an increase of 33.4% compared with the control group. The mice in the group treated with a higher dose of falcarindiol had a survival rate of 100%. Clearly, falcarindiol treatment dramatically attenuated the pathogenicity of *P. aeruginosa* and protected animals in the burned mouse model.

The results indicate that falcarindiol could attenuate *P. aeruginosa* pathogenicity in the burned mouse model (Figure 5). Considering the effect of falcarindiol on the expression of virulence genes and pathogenicity-associated phenotypical characteristics, the in vivo attenuation of *P. aeruginosa* pathogenicity was most likely due to the repression of these virulence factors. These findings indicate that falcarindiol is a promising antipathogenic drug candidate that could potentially be used for the treatment of infections caused by *P. aeruginosa* for treating *P. aeruginosa* infections.

In contrast to the conventional target of bacterial viability or growth, novel alternative targets for antimicrobial development are the functions that are essential for infection but not necessarily for viability. Bacterial virulence factors and quorum sensing systems service a function that is required for pathogens to cause host damage and disease. Antimicrobial agents targeting pathogenicity have several potential advantages over traditional antibiotics, including preserving the host endogenous microbiome and exerting less selective pressure on the pathogens. The later may theoretically result in decreased resistance [36]. 

Many studies have focused on searching for the compounds inhibiting virulence factors in *P. aeruginosa*, such as the inhibitors targeting T3SS [37,38,39], the QS system, and biofilm formation [40,41,42,43,44,45], which play crucial roles in *P. aeruginosa* pathogenicity or resistance. Falcarindiol from *Notopterygium incisum* represses multiple virulence factors in *P. aeruginosa*, including pyocyanin production, swarming motility, T3SS, elastase production, and quorum sensing, making it a promising antipathogenic compound.

The *P. aeruginosa* T3SS causes the death of many mammalian cell types in vitro, and the killing of such cells has been postulated to play an important role in the pathogenesis of pneumonia caused by *P. aeruginosa* [9,46]. Thus, T3SS has been proposed to be a promising target for treating *P. aeruginosa* infections and extensive studies subsequently have focused on the screening for the T3SS inhibitors. The vaccination against PcrV, one of the translocation proteins of effectors in T3SS, ensured the survival of challenged mice and decreased lung inflammation and injury in burned mouse models [47,48]. Additionally, a growing number of studies focus on screening small molecular inhibitors of T3SS. Some molecules inhibiting the activity of effectors have proven the effectiveness in burned mouse models [37,38], and other molecules were proven to inhibit the processes of transcription and secretion [39,49]. Apparently, T3SS is a new target for screening new inhibitors to treat *P. aeruginosa* infectious diseases. The observation that falcarindiol inhibited the expression of many T3SS genes alone suggests it is a promising drug candidate.

Pyocyanin produced by *P. aeruginosa* stimulates alveolar macrophages to produce two neutrophil chemotaxins, IL-8 and leukotriene B4, which attract neutrophils into airways to cause an inflammatory response, resulting in neutrophil-mediated tissue damages [50,51]. The wide range of biological activity associated with phenazines, such as pyocyanin, is thought to be due to their ability to undergo redox cycling in the presence of various reducing agents and oxygen, which leads to the accumulation of toxic superoxide (O_2_) and hydrogen peroxide (H_2_O_2_), and eventually to oxidative cell injury or death [52,53]. The suppression of the phenazine biosynthesis by falcarindiol would also contribute to the decreased pathogenicity.

Clearly, the attenuation of *P. aeruginosa* pathogenicity by falcarindiol was due to its synergistic effect on multiple virulence factors. However, falcarindiol is a bioactive compound isolated from a typical constituent of roots of Apiaceae plants, which have anti-inflammation activities [32,54]. It is possible that the decreased mortality in the animal studies could also be due to falcarindiol’s other bioactivities on the host.

## 3. Materials and Methods 

### 3.1. Bacterial Strains and Culture Conditions

The bacterial strains, plasmids, and gene expression reporter strains used in this study are listed in Table 1. *P. aeruginosa* PAO1 and derivatives were routinely grown at 37 °C on Lubria–Bertani (LB) agar plates or LB broth with orbital shaking at 200 rpm. Antibiotics were used at the following concentrations where appropriate: for *Escherichia coli*, kanamycin (Kn) was used at 50 μg/mL; for *P. aeruginosa*, tetracycline (Tc) was used at 300 μg/ml in *Pseudomonas* isolation agar (PIA) and trimethoprim (Tmp) was used at 300 μg/mL in LB broth. Falcarindiol (≥98%) was purchased from SenBeiJia Biological Technology Co., Ltd. (Nanjing, China), and used as indicated below. All antibiotics were purchased from MP Biomedicals LLC (Solon, OH, USA) and other chemicals were purchased from the Tianjin Kemiou Chemical Reagent Co., Ltd. (Tianjin, China).

### 3.2. Herbal Compound Crude Extraction

The air-dried herbal materials were boiled for 1.5–2 h with 95% ethanol, with the ratio of plant material weight to solvent weight set at 1:8. The crude ethanolic extracts were filtered using filter paper. The extracts were evaporated under vacuum at 40 °C using a rotary vacuum evaporator (Buchi, Switzerland) and the concentrated extracts were dried into powder and conserved at –20 °C. Extracts were reconstituted in corresponding solvent (methanol) to obtain desired dilutions for testing and filter-sterilized using 0.22 μm (pore size) filter disks.

### 3.3. Expression Monitoring and Screening for Antipathogenics

An integration plasmid CTX6.1 originating from plasmid mini-CTX-*lux* [55] was used to construct chromosomal fusion reporters. This plasmid has all the elements required for integration, the origin of replication, and a tetracycline-resistance marker. The pMS402 fragment containing the kanamycin resistance marker, the MCS, and the promoter *luxCDABE* reporter cassette was then isolated and ligated into CTX6.1. The plasmid generated was first transferred into *E. coli* SM10-λ*pir* [56] and the *P. aeruginosa* reporter integration strain was obtained using biparental mating, as reported previously [27].

The initial screening was performed on solid medium in Petri dishes inoculated with the reporter strains. Inhibitory effects of the extracts on the expression of virulence factors, pathogenicity related genes, and operons was reflected in the decreased light production of the reporter strain around the sample. Any growth effect was revealed by the zone of clearance around the samples on the plate. Briefly, overnight cultures of the reporter strains were diluted to an OD_600_ of 0.2. Aliquots of a fresh diluted culture (100 μL) were mixed with the medium (LB medium with 0.8% agar), which was cooled to 50 °C. After solidification, 10 μL of different crude extracts or solvent as control (methanol) was spotted on 6 mm diameter filter discs placed on agar plates. The plates were incubated overnight in 37 °C, and imaging was performed using a LAS3000 imaging system (Fuji Corp.). Next, liquid cultures of the reporter strains with different amounts of extracts were used to confirm the activity of virulence inhibition [57].

Virulence gene inhibition by falcarindiol was also monitored in liquid cultures using multi-well plates. Strains containing different reporters were cultivated overnight in LB broth supplemented with Tc (300 *μ*g/mL). Overnight cultures of the reporter strains were diluted to an optical density of 0.2 at 600 nm (OD_600_) and cultivated for an additional 2 h before use as inoculants. Aliquots of a fresh culture (5 μL) were inoculated into parallel wells on a 96-well black plate with a transparent bottom, which contained 95 μL of medium with or without falcarindiol stock solution (5%, vol/vol). To prevent evaporation during the assay, 70 μL of filter-sterilized mineral oil was added. Both luminescence and bacterial growth (OD_600_) were measured every 30 min for 24 h in Synergy 2 (BioTek instruments, Inc., Winooski, VT, USA). The level of light production, measured in counts per second (cps), was proportional to the level of gene expression. The light production values were then normalized to the level of bacterial growth. The level of gene expression is presented as the number of relative expression, calculated as cps/OD_600_ by use of the normalized cps values.

### 3.4. CFU Counting Using the Drop Plate Method

For counting of colony forming units (CFU), 10-fold serial dilutions were prepared in 96-well microplates using the same medium. For each dilution, 5 μL of sample was pipetted onto a plate containing agar medium with or without extracts or compounds at different concentrations. The plates were incubated until colonies were visible. All samples were used in triplicate (in three columns). The results were repeated at least three times.

### 3.5. Drosophila Melanogaster Infection Assay

The *Drosophila melanogaster* feeding assay was applied as previous reported [58]. For infection, cells from 1.0 mL of the culture were collected by centrifugation, and after removing the supernatant the pellet was resuspended in 5% sucrose with an adjusted OD_600_ of 2.0. Then, 100 μL of the resuspended cells with or without crude extract (2 mg) was spotted onto a sterile filter that was placed on the surface of 2 mL of solidified 5% sucrose agar in a 20 mL glass tube. Male flies (3–5 days old) were starved for 3 hours before 10 flies were added to each tube. Cold shock method was used to anesthetize flies throughout the sorting and transferring processes. The tubes with flies were stored at 26 °C in a humidity controlled environment. The number of live flies was counted and documented at 24 h intervals for 15 days. Four groups, including the infection group, infection with extract group, and the control groups, each had six tubes (60 flies).

### 3.6. Measurement of Pyocyanin Production

Pyocyanin was extracted from culture supernatants and measured using previously reported methods [59]. Briefly, 3 mL chloroform was added to 5 mL culture supernatant. After extraction, the chloroform layer was transferred to a fresh tube and mixed with 1 mL 0.2 M HCl. After centrifugation, the top layer (0.2 M HCl) was removed and its A_520_ was measured. The amount of pyocyanin in μg/mL was calculated using the following formula: A_520_/A_600_ × 17.072 = μg of pyocyanin per mL.

### 3.7. Swarming Motility Assays

The motility assay was carried out as described previously with minor modifications [60] . The swarming motility medium comprised 8 g/L nutrient broth, 5 g/L glucose, and 0.5% *w*/*v* agar. Swarming plates were dried at room temperature overnight before being used. Overnight cultures were spotted on the swarming plates supplemented with sample or an equal volume of solvent as control and incubated at 37 °C for 24 h. Photographs were taken with the LAS-3000 imaging system (Fuji Corp, Tokyo, Japan).

### 3.8. *Animal Studies using a Burned Mouse Model*

The experimental animals consisted of female C57BL/6J mice (weight, 20 g ± 2 g), which were purchased from the Experimental Animal Center of the Fourth Military Medical University, Xi’an, Shaanxi, China. The animals were housed in ventilated cages in a pathogen-free facility operated with 12 h light/dark cycles at 23 °C (standard deviation [SD], 2 °C) and 30% to 60% humidity.

The mice were anesthetized and shaved on their back. All mice were burned about 2 cm in diameter throughout the full thickness of skin using 95 to 98 °C water steam for 10 s. When the burned regions were cooled down for about 5 min to normal body temperature, the surface area of treatment groups was inoculated with 50 μL of *P. aeruginosa* bacterial suspension using a microinjector. Subsequently, all mice were administered intraperitoneally with 1 mL of physiological saline immediately afterwards. The mice were monitored daily for their survival up to 6–8 days after burn injury.

For the initial pilot study, two groups of mice were used, with 10 mice per group. The *P. aeruginosa* concentration in the inoculant suspension used on the burn eschar was 1 × 10^9^ CFU/mL. The mice were randomly divided into treatment group and control group (saline containing 5% ethanol). Administration of drugs or saline containing 5% ethanol (as a control) started at 12 h postinfection. Individual animals in the treatment group were administered intraperitoneally with 400 μL of falcarindiol at a final concentration of 500 μg/mL in saline containing 5% ethanol to provide a dosage of 10 mg/kg. The control group was injected with 400 μL of saline containing 5% ethanol. The administration of drugs and saline containing 5% ethanol was carried out once a day for 6 days. The number of dead animals was documented daily and mortality was calculated.

During the second experiment, three groups of mice were used, with 12 mice per group, and the *P. aeruginosa* concentration in the inoculant suspension used on the burn eschar was adjusted to 1 × 10^7^ CFU/mL. The mice were randomly divided into two treatment groups (high dose and low dose groups) and a control group. Administration of drugs or control solution (saline containing 5% ethanol) started 12 h postinfection as before. Individual animals in the high and low dose treatment groups were given falcarindiol at 10 mg/kg and 6 mg/kg respectively. The administration of drugs and control solution was carried out once a day for 8 days.

### 3.9. Statistical Analysis

Student’s *t* test was used to analyze the data. Survival data were analyzed using Kaplan–Meier survival curves with GraphPad Prism 5.0 (GraphPad Software Inc., La Jolla, CA, USA) and significance was examined by log rank (Mantel–Cox) analysis.

## 4. Conclusions

Our data indicate that falcarindiol from *Notopterygium incisum* inhibits multiple virulence factors in *P. aeruginosa* and decreases the mortality of burned mice infected with *P. aeruginosa*. Falcarindiol represents a promising novel antipathogenic drug candidate for treating *P. aeruginosa* infections.

## Figures and Tables

**Figure 1 antibiotics-09-00136-f001:**
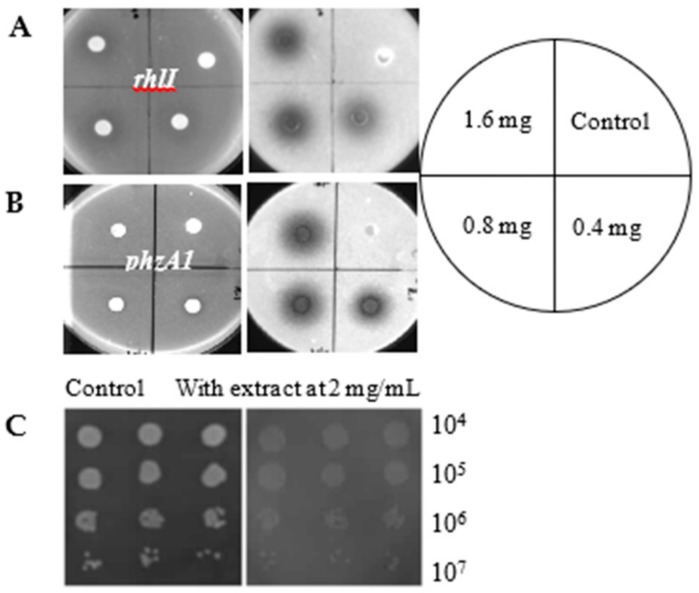
Effect of *Notopterygium incisum* crude extract on *rhlI* (**A**) and *phzA1* (**B**) expression by disc diffusion assay. Reporter strains were mixed with the Luria–Bertani (LB) agar medium (Luria–Bertani broth with 0.8% agar), and after incubation, images of the plates were captured under both white light (left) and in dark (right). In dark, inhibition of light production of the reporters is visualized as dark zones around the samples. The concentrations of crude extracts used in each spot are indicated on the right. (**C**) The growth comparison by CFU counting using drop plate method. The grayish color on the right was due to the color of the crude extract. The samples were tested in triplicate in three columns.

**Figure 2 antibiotics-09-00136-f002:**
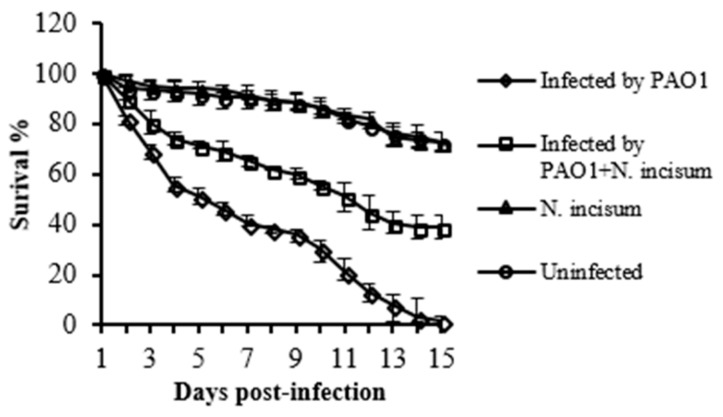
The effect of crude extract from *Notopterygium incisum* on *Drosophila melanogaster* infection. The fly feeding assay was used. Survival curves of flies infected with PAO1 alone (diamonds) or together with crude extract (squares) are shown. Uninfected subjects (circles) and feeding with *Notopterygium incisum* without infection (triangles) were the controls.

**Figure 3 antibiotics-09-00136-f003:**
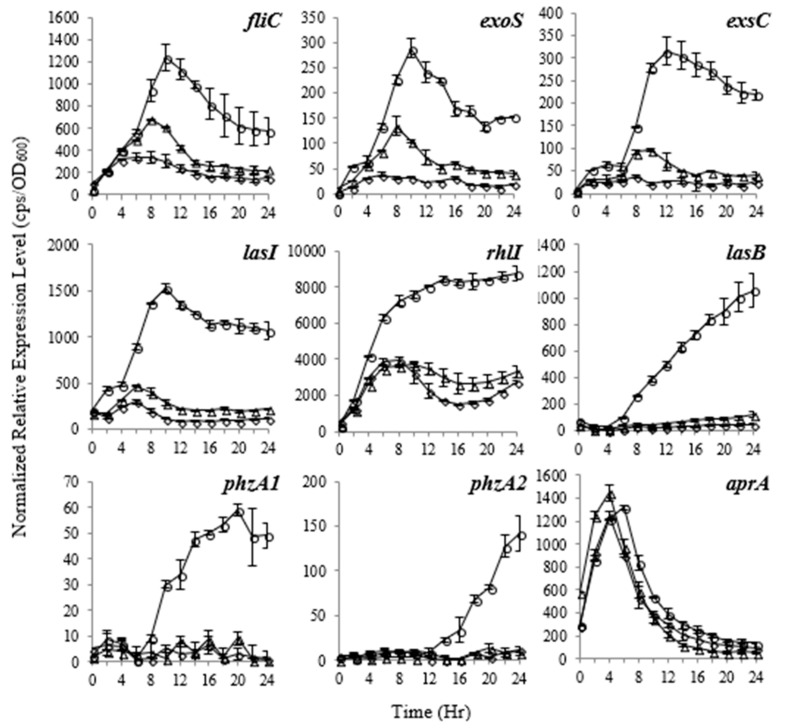
The expression profiles of virulence factors with different concentrations of falcarindiol. The promoter activities were measured using p-lux promoter–reporter system. The relative expression values are presented as cps normalized to OD_600_. **Circles**: without falcarindiol; **triangles**: promoter activities at 8 μg/ml falcarindiol; **diamonds**: promoter activities at 32 μg/ml falcarindiol. No effect was observed on virulence factor gene *aprA*. The experiments were repeated at least three times. The means and standard deviations are presented.

**Figure 4 antibiotics-09-00136-f004:**
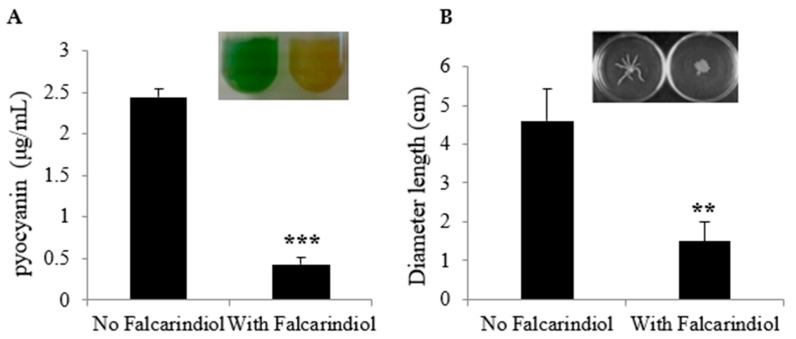
The effect of falcarindiol on pyocyanin production (**A**) and swarming motility (**B**). Pyocyanin production and swarming motility were measured in the presence of 32 μg/mL falcarindiol. Note: ****p*<0.001, ** *p* <0.01. The images of the cultures and swarming motility tests are shown as inserts.

**Figure 5 antibiotics-09-00136-f005:**
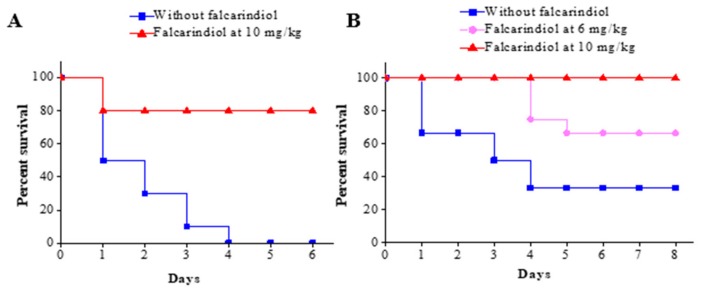
Attenuation of *P. aeruginosa* pathogenicity by falcarindiol. Kaplan–Meier survival curves of the burned mouse model are plotted with high infection dose (5×10^7^ CFU) of *P. aeruginosa* (**A**) and lower infection dose (5×10^5^ CFU) (**B**). Mice in the control groups (without falcarindiol) were infected but treated with saline containing 5% ethanol. Log rank analysis (Mantel–Cox) was used to compare the survival rates between the groups. Significant differences were observed between the two groups in the first experiment (n=10 per group) (**A**) (*p* < 0.0001), and between the control and both the 6 mg/kg (*p* < 0.05) and the 10 mg/kg (*p* < 0.005) groups in the second experiment (n=12 per group) (**B**).

**Table 1 antibiotics-09-00136-t001:** Bacterial strains and plasmids used in this study.

Strains or Plasmids	Relevant Characteristics	Source or Reference
**Strains**		
PAO1	Wild-type	[26]
*E. coli* SM10-λ*pir*	mobilizing strain, RP4 integrated in the chromosome; Kn^r^	Invitrogen
CTX- *fliG*	Integration reporter strain, CTX6.1 with a fragment of pKD-*fliG* containing *fliG* promoter region and *luxCDABE* gene; Kn^r^, Tmp^r^, Tc^r^	This study
CTX- *fliC*	Integration reporter strain, CTX6.1 with a fragment of pKD-*fliC* containing *fliC* promoter region and *luxCDABE* gene; Kn^r^, Tmp^r^, Tc^r^	This study
CTX-*phzA1*	Integration reporter strain, CTX6.1 with a fragment of pKD-*phzA1* containing *phzA1* promoter region and *luxCDABE* gene; Kn^r^, Tmp^r^, Tc^r^	[23]
CTX-*phzA2*	Integration reporter strain, CTX6.1 with a fragment of pKD-*phzA2* containing *phzA2* promoter region and *luxCDABE* gene; Kn^r^, Tmp^r^, Tc^r^	[23]
CTX-*exoS*	Integration reporter strain, CTX6.1 with a fragment of pKD-*exoS* containing *exoS* promoter region and *luxCDABE* gene; Kn^r^, Tmp^r^, Tc^r^	[23]
CTX-*exoT*	Integration reporter strain, CTX6.1 with a fragment of pKD-*exoT* containing *exoT* promoter region and *luxCDABE* gene; Kn^r^, Tmp^r^, Tc^r^	[23]
CTX-*exsD*	Integration reporter strain, CTX6.1 with a fragment of pKD-*exsD* containing *exsD* promoter region and *luxCDABE* gene; Kn^r^, Tmp^r^, Tc^r^	[25]
CTX-*exsC*	Integration reporter strain, CTX6.1 with a fragment of pKD-*exsC* containing *exsC* promoter region and *luxCDABE* gene; Kn^r^, Tmp^r^, Tc^r^	[25]
CTX-*lasI*	Integration reporter strain, CTX6.1 with a fragment of pKD-*lasI* containing *lasI* promoter region and *luxCDABE* gene; Kn^r^, Tmp^r^, Tc^r^	This study
CTX-*lasB*	Integration reporter strain, CTX6.1 with a fragment of pKD-*lasB* containing *lasB* promoter region and *luxCDABE* gene; Kn^r^, Tmp^r^, Tc^r^	This study
CTX-*lasR*	Integration reporter strain, CTX6.1 with a fragment of pKD-*lasR* containing *lasR* promoter region and *luxCDABE* gene; Kn^r^, Tmp^r^, Tc^r^	This study
CTX-*rhlI*	Integration reporter strain, CTX6.1 with a fragment of pKD-*rhlI* containing *rhlI* promoter region and *luxCDABE* gene; Kn^r^, Tmp^r^, Tc^r^	This study
CTX-*rhlR*	Integration reporter strain, CTX6.1 with a fragment of pKD-*rhlR* containing *rhlR* promoter region and *luxCDABE* gene; Kn^r^, Tmp^r^, Tc^r^	This study
CTX-*aprA*	Integration reporter strain, CTX6.1 with a fragment of pKD-*aprA* containing *aprA* promoter region and *luxCDABE* gene; Kn^r^, Tmp^r^, Tc^r^	This study
Plasmids		
pMS402	Expression reporter plasmid carrying the promoterless *luxCDABE* gene; Kn^r^, Tmp^r^	[22]
CTX6.1	Integration plasmid origins of plasmid mini-CTX-*lux*; Tc^r^	[25]
pRK2013	Broad-host-range helper vector; Tra^+^, Kn^r^	[27]

Kn^r^, Kanamycin resistance; Tmp^r^, Trimethoprim resistance; Tc^r^, Tetracycline resistance.

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
