# Peer review of "Virulence-Inhibiting Herbal Compound Falcarindiol Significantly Reduced Mortality in Mice Infected with Pseudomonas aeruginosa"

_antibiotics, 2020, doi:10.3390/antibiotics9030136_

Round 1

Reviewer 1 Report

The authors in the manuscript titled " Virulence-inhibiting herbal compound falcarindiol significantly reduced mortality in mice infected with Pseudomonas aeruginosa" have successfully shown that falcarindiol from Notopterygium incisum inhibits multiple virulence 336 factors in P. aeruginosa and decreases the mortality of burned mice infected with P. aeruginosa. It is overall a good manuscript, the experiemnts have been well thought and carefully executed. 

Few concerns that the authors need to address is how many times the experiments were repeated, how many mice were included in the in vivo study, and explain in detail how the activity of the crude extraction of the compound was detected? 

I would also suggest the authors to proof read the manuscript for grammatical errors. 

Reviewer 2 Report

Dear colleagues,

In this work you have worked on an herbal extract. You focused on its ability to alter the virulence of P. aeruginosa. During a primary approach you used transcriptional fusions to assess the effect of the extract on the transcription of several genes involved in the virulence of P. aeruginosa. According to the material and method part these experiments were conducted on Petri dishes as well as on microplates. Afterwards the extract was tested on an infection model (Drosophila melanogaster infection).

The study was then pursued on one molecule present within the extract the falcarindiol. The molecule was tested again on several transcriptional fusions to assess its effect on the transcription of several genes involved in the virulence of P. aeruginosa. It was further demonstrated that this molecule could reduce the production of phenazine and the swarming motility of P. aeruginosa. The last experiments were conducterd on a mice infection model were the falcarindiol increase the survival rate of mice infected by P. aeruginosa.

I have found two typing mistakes:

L 82: “Results and discussion”

L 120: “The next approach WORLD be to isolate…”

Recommendations:

You should be careful when dealing with the presentation of the transcriptional fusions results: “The results indicate that there are bioactive compounds in the Notopterygium incisum extract that could inhibit P. aeruginosa virulence factors”: instead of this, I would rather prefer something like “The results indicate that there are bioactive compounds in the Notopterygium incisum extract that could inhibit the transcription of genes involved in the virulence of P. aeruginosa.” You effectively established inhibithion  the virulence factors during the phenazine dosage and the swarming motility assay, but when you used the transcriptional fusions it was preliminary and needed to be further tested.

I am quite skeptical regarding the assertion: “At the concentration used in the experiments, no inhibition of P. aeruginosa growth by the extract was observed”. Apparently, you used OD600 measurements but never bacterial cell counting (CFU) ? With OD600 you only see the shadow of the micro-organisms… they can be alive or dead… alive or in dormancy… you can’t say.

Without these countings you can’t affirm that you have found a molecule that targets only the virulence and not the growth. In fact, it could alter the growth very lightly, just enough to decrease all the metabolism of P. aeruginosa and therefore its production of virulence factors: it won't be a specific anti-virulence activity on the contrary of the halogenated furanones of Hentzer and colleagues…

best wishes

Round 2

Reviewer 2 Report

Dear colleague,

thank you very much for your work on the article. Thank you also for your explanations. 

have a nice day